# How Well Do Dogs Cope with Air Travel? An Owner-Reported Survey Study

**DOI:** 10.3390/ani13193093

**Published:** 2023-10-04

**Authors:** Katrin Jahn, Jacqui Ley, Theresa DePorter, Kersti Seksel

**Affiliations:** 1German Veterinary Clinic, Villa 112, 39th Street, Khalifa City A, Abu Dhabi P.O. Box 34867, United Arab Emirates; 2Melbourne Veterinary Specialists Centre, 70 Blackburn Rd, Glen Waverley, VIC 3150, Australia; behaviour@melbvet.com.au; 3Oakland Veterinary Referral Services, 1400 S Telegraph Road, Bloomfield Hills, MI 48302, USA; 4Kersti Seksel and Associates Ltd., Seaforth, NSW 2092, Australia

**Keywords:** air travel, travel, stress management, welfare, human–animal bond, anxiolytics, pheromones

## Abstract

**Simple Summary:**

Pet air travel has increased in the last decade, and 6% of pets in the US board a plane every year. Dogs have been reported to make up 58% of pets travelling worldwide. Despite this, there is little data available about air travel in dogs. A total of 635 questionnaires were collected from dog owners whose dogs had travelled by air in the last 12 months to obtain initial data on international dog air travel and how well dogs cope with air travel physically, mentally, and emotionally. Data on how dogs were being prepared for air travel, specifically regarding stress management, were also collected. Results showed that most dogs cope with and recover well from air travel but that there is a group of individuals who suffer physical, mental, and emotional ill health consequences during or after air travel, including death. Most dog owners planned air travel themselves, and over half did not seek professional advice. Stress management products such as anxiolytic medication and supplements or pheromones were only used in one-quarter of canine travellers. This study presents opportunities for all stakeholders of pet air travel, including owners, veterinarians, airlines, airports, and pet shippers, to improve pet welfare during air travel.

**Abstract:**

It is estimated that 2 million domestic animals travel on commercial flights every year in the US alone and that dogs make up 58% of pets travelling worldwide. There has been little research on the welfare effects of air travel on dogs. The purpose of this owner-reported study was to understand how well dogs cope with and recover from air travel from a physical, mental, and emotional health perspective. An online survey questionnaire was distributed globally to pet owners whose dogs had travelled by air in the last 12 months, and the results were collected and analysed. Information was received about dog and owner demographics, logistics, and preparation for travel, as well as the dog’s experience of air travel. Results showed that most dogs cope with and recover well from air travel but that there is a group of individuals who suffer physical, mental, and emotional ill health consequences during or after air travel, including death. Stress management products such as anxiolytic medication, supplements, and pheromones were underutilised and, in some instances, actively discouraged. More education of all stakeholders of pet air travel is needed to improve the physical, mental, and emotional health and welfare of canine air travellers.

## 1. Introduction

Pet travel has increased by 19% in the last decade [1], and it is estimated that 2 million domestic animals travel on commercial flights every year in the US alone [2]. Moreover, 6% of pets in the US board a plane every year, and dogs have been reported to make up 58% of pets travelling worldwide [2]. Considering these large numbers, there is generally little data available about air travel in dogs and specifically how dogs cope with air travel, and there have been only a few publications looking at air travel and stress in dogs. One study [3] investigating the physiological signs and behaviour of 24 Beagles during air transport concluded that air transportation is stressful for dogs and that sedation with acepromazine did not affect physiological and behavioural stress responses. A further study [4] looked at the relationship between kennel size and stress in 12 greyhounds travelling between Ireland and England by air and concluded that some greyhounds found the experience extremely stressful regardless of kennel size. Studies in other species show that horses experience a sharp increase in heart rate and changes in behavioural activities during air transport, especially during the transitional stages such as the aircraft ascending and descending [5] and in two Giant Pandas transported by air from China to the US, their urinary cortisol measures were highest during the time of the flight compared to the remainder of the 30-day period post-transport [6].

Regarding animal deaths, injuries, or loss on flights, data are easily accessible only from the Department of Transportation (DOT) in the US. The US DOT Air Travel Consumer Report for 2022 shows 7 animal deaths, injuries to 1 other animal, and 1 lost animal, for a total of 9 incidents, down from the 19 incident reports filed for the pre-pandemic calendar year 2019. For the calendar year 2022, 188,223 animals were transported by US airlines, for a rate of 0.48 incidents per 10,000 animals transported [7]. In pre-pandemic 2019, 404,556 animals were transported, for a rate of 0.47 incidents per 10,000 animals transported [8]. While this seems like a low incidence, it is important to consider that this report applies only to US carriers, is not a reflection of the global situation and that incidents occurring immediately after a flight, for example, in quarantine facilities, are not accounted for.

The purpose of this survey study was to collect some initial data on international dog air travel and to obtain information on how well dogs cope with air travel from a physical, mental, and emotional health perspective. Furthermore, we wanted to collect data pertaining to how dogs were being prepared for air travel and what measures were commonly being taken to manage their stress during air travel. It is our hypothesis that most dogs cope well with and recover quickly from air travel but that there are a certain number that suffer physical, mental, and emotional ill health consequences during and after air travel. We also hypothesise that dog owners worry about their dogs traveling by air and that, currently, stress management practices for dogs are underutilised or even actively discouraged.

## 2. Materials and Methods

### 2.1. Subjects

The inclusion criteria for participants of this web survey were dog owners from any country around the world whose dog had travelled by air in the last 12 months. We chose the timeframe of 12 months as the questionnaire asked several descriptive questions relating to the dog’s air travel experience, and we wanted to ensure good recollection of this on the part of the participant whilst giving a large enough timeframe to capture a representative number of participants. If the participant had more than one dog who had travelled by air in the last 12 months, we asked them to choose the dog whose name started closest to the letter “A”.

### 2.2. Survey Questionnaire and Data Collection

A web-based, electronic survey was designed in English on Survey Monkey© and initially circulated to a pilot cohort of 18 people to test functionality and ease of completion. Feedback was collected from the pilot cohort, and some minor changes were made to the survey based on this feedback.

The survey comprised 51 questions divided into three sections:(1)**Demographic data—dog**: This section included questions about the dog’s age, gender and neuter status, body weight and size, breed, hair coat, whether they were brachycephalic, age at which and how the dog was acquired, purpose of having the dog, physical and mental/emotional health, training history and previous flight experience.(2)**Demographic data—owner**: This section included questions about the owner’s age, gender, education, nationality, country of residence, and how stressed they felt at the thought of their dog travelling by air.(3)**Data about the dog’s air travel process and experience**: This was the most comprehensive section and was further divided into questions about
(a)Logistics and preparation for air travel;(b)The dog’s experience of air travel.

This section included questions about the reason for air travel, the dog’s age at the time of air travel, the route and length of air travel, the number of transit stops during air travel, where in the plane the dog travelled (for example, in the cabin or the hold of the aircraft), preparation for air travel, signs of stress/distress at different times during the air travel process and physical, mental, and emotional health after air travel.

The questions took a variety of forms, including multiple choice questions (e.g., the dog’s age), ticking checklists (e.g., how owners prepared their dog for air travel), open-ended questions (e.g., which country the dog’s flight started from), and Likert scale questions (e.g., how stressed the owner felt prior to their dog travelling by air). An “other” option was included wherever appropriate to allow participants to provide further information. The amount of time needed for completion was estimated at 15–20 min. The full survey questionnaire is posted in the Appendix A.

In the introduction of the survey, participants were informed that their participation was completely voluntary and that they were able to stop the survey and their participation at any time if they wished. They were further informed that the survey was anonymous and that they could not be identified during any stage of the process. Consent was received from participants for data being collected and stored as a result of their participation in the survey and for this data to be published. The primary author’s email address was provided as a way of contact and for any questions or concerns from participants, and participants were given the option for their data to be withdrawn at any point by contacting the same email address.

The survey was posted on various Facebook groups (including dog owner, dog breeder, pet travel, pet air travel, and global expatriate Facebook groups) as well as the primary author’s Instagram and LinkedIn pages entitled “How well did your dog cope with air travel?” and sent via email to pet shippers belonging to the IPATA (International Pet and Animal Transportation Association) organisation for circulating amongst their client lists from November 2022 to February 2023. To further generate participation, we encouraged people to share links to the survey on their own social media accounts and to tag people whose dogs had travelled by air in the last 12 months. Additionally, dog owners who had completed the questionnaire were eligible to receive a free PDF designed by the first author entitled “Top tips for air travel with pets”. To maintain anonymity, participants were asked to email the first author directly or share their email address on the Facebook Messenger application to receive the PDF.

### 2.3. Statistical Analysis

A total of 771 surveys were received; of these, 136 were discarded due to not being fully completed or being completed for a different species (cat), and a total of 635 questionnaires were statistically analysed. Data collected from the online questionnaires were initially downloaded into a Microsoft Excel spreadsheet, where it was coded. Descriptive statistics were performed using the statistics program IBM SPSS Statistics Version 29. All the answers for each criterion were plotted, and the percentage was calculated for each category.

## 3. Results

### 3.1. Demographic Data—Dog

At the time of air travel and survey completion, most dogs were adults between 2 and 8 years of age. At the time of air travel, only 10.1% were under 6 months of age, and 16.4% were over 8 years of age (see Table 1).

Most dogs (59.1%) were acquired as puppies at the age of 8–16 weeks, and 20.9% of dogs were acquired at the age of 4–24 months. Just under half the dogs in the survey were acquired from private breeders, and around one-quarter of dogs in the survey were acquired from a shelter or rescue (See Figure 1).

A total of 80% of dogs in the survey were acquired as companions and around 25% as family dogs. Approximately 25% each were acquired for breeding purposes or as sports dogs.

At the time of survey completion, most dogs were neutered (67.4%), with an almost equal distribution of male and female dogs.

As shown in Figure 2, most dogs were medium-sized dogs (10–25 kg bodyweight) or small dogs (5–10 kg). Moreover, 19.7% were large dogs (25–40 kg), 13.9% were miniature dogs (under 5 kg), and only 2% were giant breed dogs (over 40 kg).

A total of 58.6% were purebred dogs, with the most common breed groups (according to the seven UK Kennel Club breed groups) being pastoral—also known as working- or herding breeds—(17.8%) and toy (17.6%) breeds (see Figure 3). Hybrid breeds are defined in this survey as modern designer breed dogs such as Labradoodles and Cockapoos and made up 5.8% of dogs in this study. Moreover, 8.3% (53 dogs) were brachycephalic breed dogs.

Most dogs had either short or medium hair coats, 11.5% had long hair coats, and 9.4% had dense or thick hair coats.

Moreover, 67.4% of dogs did not have any owner-reported physical illnesses at the time of the flight, and where owners did report physical illnesses, 13.2% of dogs suffered from food- or environmental allergies, and 7.6% of dogs suffered from osteoarthritis or musculoskeletal disease (see Table 2).

Moreover, 44.3% of owners reported that their dog suffered from a mental/emotional illness at the time of the flight. In addition, 14.6% reported that their dog suffered from separation anxiety, and 13.5% reported that their dog suffered from more than one of the behaviour disorders mentioned for selection in the questionnaire (see Table 3).

Most dogs (81.1%) were not taking any regular medications at the time of air travel, and of the dogs taking medication, 5.5% were taking psychotropic medication, 3.3% were taking non-prescription supplements, 3.1% were taking regular parasite control medications, 2.2% were taking allergy control medications, and 1.9% were taking non-steroidal anti-inflammatory medications.

The distribution of frequency of air travel of the dogs in this survey is shown in Figure 4.

### 3.2. Demographic Data—Owner

A total of 90.6% of participants were female, and 78% of participants were between the ages of 25 and 54.

Moreover, 63.5% of participants held either an undergraduate or postgraduate degree.

A total of 49% of participants resided in North America, 26.8% in Europe, and 11.2% in the Middle East/Africa region. Moreover, 5.7% resided in Asia, 4.4% in Oceania (Australia, New Zealand, and Pacific Islands), and 2.8% in Latin America. The nationality of participants was as follows: 48% were North American, 28.7% European, between 4 and 5% of participants were Asian or from Oceania, respectively, or had dual nationality, and between 3 and 4% of participants, respectively, were of the Middle East/African or Latin American nationalities, and 2.7% preferred not to answer this question.

When asked how stressed the participants felt at the thought of their dog travelling by air, 41.4% were either extremely stressed or very stressed, 34% were somewhat stressed, 17.5% were not very stressed, and only 7.1% were not at all stressed (see Figure 5).

### 3.3. Data about the Dog’s Air Travel Process and Experience

#### 3.3.1. Logistics and Preparation for Air Travel

The primary reason for dogs travelling by air in this survey was relocation (43.5%). This was followed by taking the dog on vacation (24.3%). Moreover, 8.5% of participants received their dog via air transportation as puppies, 7.9% of dogs travelled as service dogs, and 6.9% of dogs travelled to dog shows or competitions (see Figure 6).

Most flights both originated (52.8%) and terminated (44.4%) in North America, 22% originated and 31% terminated in Europe, 9.1% originated and 10.1% terminated in the Middle East/Africa and 8.5% originated and 5.4% terminated in Asia.

Most flights (60.9%) were direct flights, 33.1% of flights had 1 stop/layover, and 6% of flights had multiple stops/layovers.

35.4% of trips were termed long trips (10–24 h), 26.1% were medium-length trips (6–10 h), 24.3% were short trips (1–6 h), 9.4% were multi-leg with multiple stopovers, and only 4.7% were long/multi-leg trips with boarding and/or quarantine stays.

Participants were asked where their dogs were cared for during different stages of the flight and were asked specifically about dedicated airport animal lounges, where dogs are kept in separated, temperature-controlled environments, are possibly taken out of their crates and offered food and water, and possibly examined by a veterinarian or non-veterinary care assistant. Dedicated airport animal lounges, if present, are usually part of the cargo shipment area and would not be used by dogs travelling in the cabin. A total of 8.4% said that their dogs did not visit a dedicated airport animal lounge, 24.9% of participants said that yes, their dog was cared for at a dedicated airport animal lounge at least at one point of air travel and 16.7% of participants did not know what a dedicated airport animal lounge was or did not know where their dog was cared for during air travel.

A total of 51.2% of dogs travelled in the hold of the aircraft, and 48.5% of dogs travelled in the cabin, either on a leash or in a carrier and either with the participant or another accompanying person.

A total of 13 dogs (2%) were said to be emotional support dogs, and of these, 11 travelled in the cabin with the participant.

Moreover, 23.1% of participants used a pet shipping company to plan the entire process of pet air travel, 66.8% planned and prepared the trip completely by themselves, and 10.1% used a pet shipping company for some aspects of the trip preparation and planned other aspects themselves.

Most participants (21.6%) started preparing their dogs for air travel 1–4 weeks before the travel date, with the second largest group (18.1%) beginning preparation 1–3 months before travel. A smaller number of owners started preparing more than 6 months before travel (14.3%), and surprisingly, 13.9% only started preparing their dogs 24 h to 1 week before travel. Around 10% each started 3–6 months before travel, 1 day before travel, or did not prepare their dogs for travel themselves (see Figure 7).

Over half of the participants (55.3%) stated that they did not seek any advice from professionals, and 6.5% used their own previous experience to prepare their dogs for air travel.

Of the participants that sought advice from professionals, 39.5% sought advice from airlines, 37.3% from IATA (International Air Transport Association), 31.8% from books, 29.8% from veterinarians, 24.9% from the Internet/Google and 15.9% from pet shippers. Advice was sought to a lesser degree from dog trainers, other pet professionals, family members, pet shipping organisations such as IPATA (International Pet and Animal Transportation Association) and ATA (Animal Transportation Association), social media, breeders, and government agencies (see Table 4).

When asked about the preparation of their dog for air travel, 67.4% of participants crate-trained their dogs prior to air travel, 74.8% visited a veterinarian to perform a physical examination on their dog, and 79.2% visited a veterinarian to perform required vaccinations, blood tests, microchip placement or parasite treatments prior to air travel.

When asked about the use of stress management or calming products, 13.4% of participants used anxiolytic medications, 12.8% used calming supplements, 9.8% used pheromone products, and 2% used a calming diet in preparation for or during the flight. A total of 5.8% of participants used CBD/hemp products, and 3.1% used aromatherapy products. A total of 76.7% of participants did not use any kind of stress management products (medications, pheromones, supplements, and diets), and seven participants (1.1%) were pro-actively advised not to use any kinds of calming or anxiolytic products before or during air travel. When anxiolytic medication or supplements were used, 49 participants (7.7%) used trazodone, 34 (5.4%) used gabapentin, 11 (1.7%) used alpha-casozepine (Zylkene^®^, Vetoquinol—Lure, France), 10 (1.6%) used a benzodiazepine, 6 (0.9%) each used YuCalm^®^ (Lintbells—Hitchin, UK) and the anti-nausea medication maropitant, 3 (0.5%) each used acepromazine and Benadryl^®^ and even fewer used clonidine, L- Theanine (Anxitane^®^, Virbac—Carros, France), Composure^®^ (VetriScience—VT, USA) and Soliquin^®^ (Nutramax—SC, USA). No participants used oromucosal dexmedetomidine gel (Sileo^®^, Zoetis—NJ, USA) or pregabalin.

A total of 39 participants (6.1%) used behaviour modification techniques such as desensitisation and familiarisation with certain aspects of the air travel process, four participants put familiar items in the dog’s crate, and three participants used a Thundershirt^®^ during air travel.

Other methods of preparation included the dog losing weight (1 participant), putting a tile tracker (a Bluetooth device that can be attached to an object to track its location) on the dog’s collar (1 participant), and chartering a private jet for air travel (2 participants).

#### 3.3.2. The Dog’s Experience of Air Travel

Participants were asked how distressed they felt their dogs were at the following points during air travel: 1. When handed over to the pet shipper or airport staff before the flight (if the dog travelled in the hold of the aircraft), 2. At the airport or during the flight (if the dog flew in a cabin with the owner or an accompanying person), 3. At the time of arrival/collection at the airport or when the dog was delivered to the participant after the flight, the results can be seen in Table 5.

Participants were also asked whether they saw their dog displaying certain body language signs associated with stress at the three points during air travel: (1) when handed over to the pet shipper or airport staff before the flight (if the dog travelled in the hold of the aircraft), (2) at the airport or during the flight (if the dog flew in a cabin with the owner or an accompanying person), and (3) at the time of arrival/collection at the airport or when the dog was delivered to the participant after the flight (see Table 6).

A total of 70.1% of dogs showed at least one sign of the above-mentioned stress signs at handover before the flight, 72.4% of dogs showed at least one of the above-mentioned stress signs during the flight, and 32% of dogs showed at least one of the above-mentioned signs of stress at arrival/handover after the flight.

Moreover, 3% of respondents stated that their dog had shown “other” signs of stress before, 4.1% during and 4.9% after the flight (open-ended question box). Some of these responses included being wide-eyed, the tail being tucked, being restless, unable to settle and hyper-alert to the surroundings, biting on the water bowl and mesh covering of the travel crate, chewing the leash tied onto the crate to pieces, having a raw nose from rubbing it on the crate door, being desperate to urinate and defecate, being thirsty, being tired, not wanting to eat food or treats, not moving, having trouble standing and walking, trying to escape from crate, pulling on the leash whilst walking through the airport, lunging from inside the crate when approached, and, sadly, one dog died in quarantine after the flight.

Most dogs (83.6%) did not receive medical attention from a veterinarian immediately after air travel. Of the dogs that did receive medical attention after the flight, 83 dogs (13.1%) received a routine veterinary examination, 18 dogs were seen unrelated to air travel (for example, to refill regular medication), 8 dogs were seen to treat a major health issue, 1 dog was seen to treat a minor health issue, 1 dog was seen to treat a mental/emotional/behavioural condition, and 2 dogs were seen to confirm death.

Participants were asked to comment on the dog’s stress level at various times following air travel (48 h, 3–7 days, 8–30 days, and over 30 days after air travel), and the responses can be seen in Table 7.

Participants were asked about the dog’s behaviours and signs of stress at various times following air travel (48 h, 3–7 days, 8–30 days, over 30 days). Between 34% (2 days after air travel) and 39.4% (over 30 days after air travel) of participants stated that their dogs did not show any behaviours or signs associated with stress after air travel. Of the dogs that did show behaviours or signs associated with stress, the most commonly seen are listed in Table 8.

A total of 30 participants (4.7%) responded that their dog had shown “other” behaviours or signs of stress after air travel. These included howling and crying when being left alone, especially for longer periods, being more alert and jumpier, being unwilling to go for a walk, being “overheated”, vomiting, and being unsettled with another dog in the household.

A total of 13.8% of participants stated that their dog developed a behaviour problem within 3 months after the flight. Moreover, 4.3% of dogs developed separation anxiety, 3% of dogs were generally more anxious, and 3.6% of dogs developed more than one of the behaviour disorders mentioned for selection in the questionnaire (see Table 9).

A total of 8.2% of participants thought that the behaviour change was a result of the flight, 12.1% thought it was due to a change in the dog’s physical environment, and 9% thought it was due to a change in the dog’s social environment (more than one answer was possible).

A total of 10.4% of participants stated that their dog experienced worsening of an already existing behaviour problem within 3 months after the flight. Moreover, 4.1% of dogs experienced a worsening in separation anxiety, 1.1% experienced a worsening in aggressive behaviours, and 3.6% of dogs experienced a worsening of more than one of the behaviour disorders mentioned for selection in the questionnaire (see Table 10).

A total of 5.8% of participants thought that the worsening of the behaviour problem/s was a result of the flight, 7.4% thought it was due to a change in the dog’s physical environment, and 6.3% thought it was due to a change in the dog’s social environment (more than one answer was possible).

A total of 8.3% of participants stated that their dog experienced worsening of an already existing physical illness within 3 months after the flight. Moreover, 2.7% of dogs experienced a worsening in food or environmental allergies, 1.9% experienced a worsening in osteoarthritis or musculoskeletal disease, and 1.3% of dogs experienced a worsening in gastro-intestinal disease (see Table 11).

A total of 2.8% of participants thought that the worsening of the physical illness was a result of the flight, 5.7% thought it was due to a change in the dog’s physical environment, and 2.2% thought it was due to a change in the dog’s social environment (more than one answer was possible).

### 3.4. The Two Dogs That Died

Questionnaire 28 was a 4–6-year-old female neutered, ex-racing greyhound weighing 25–40 kg. She was obtained from a shelter between 4–24 months of age. There were no known pre-existing physical medical illnesses; however, her owner–reported pre-existing mental/emotional illnesses, namely separation anxiety and fears and phobias of specific triggers. She was not taking any medication at the time of the flight and had never travelled by air before. Her owner was very stressed about the thought of her dog travelling and used a pet shipping company to arrange the entire travel process.

Travel was to be from the UK to Australia (a long, multi-leg trip including quarantine), with one transit in Singapore; however, the dog did not make it to Australia and died in the quarantine facility in Singapore. The dog travelled in the hold of the aircraft.

In preparation for travel, the dog underwent a physical examination by a veterinarian and received vaccinations and blood tests for travel. No further preparation or stress management was performed, and the owner was told that “dogs are not allowed any calming medications” for air travel, which was advice received from a veterinarian and the pet shipping agent.

The dog was very distressed at handover, and behavioural signs of stress included trembling, panting, cowering, hiding, being tense, scratching, and “shake off”.

Questionnaire 330 was a 4–6-year-old male entire, French Bulldog of medium size (10–25 kg). He was obtained at 8–16 weeks of age from an animal farm/shop in Dubai as a family companion and for his breed and look/aesthetics. He had a known owner-reported pre-existing physical disease in the form of skin allergies and food hypersensitivities but suffered from no known mental/emotional disease and was not taking any medications. He had travelled by air five times before.

His owner was extremely stressed at the thought of her dog travelling and used a pet shipping company to arrange the entire travel process. Travel was for the purpose of relocating to live in another country, and the route was from Spain to Singapore (long trip—10–24 h) with multiple transits/stops. The dog flew in a “special animal freight plane”, and the owner does not know if this was a cargo or passenger plane or if the dog flew in the hold or cabin of the aircraft.

In preparation for travel, the dog underwent a physical examination by a veterinarian and received vaccinations and blood tests for travel. Preparation started 3–6 months prior to travel and included travel crate familiarisation. No further preparation or stress management was performed.

The dog was reported to be “not so distressed” at handover before the flight, and sadly, he was found dead on arrival in his crate in Singapore.

## 4. Discussion

### 4.1. Subjects and Data Collection

The validity of behavioural survey reporting by pet owners as a form of observational study has been confirmed in several papers, and questionnaires are a commonly used method of collecting behavioural data from companion animals [9]. However, despite their frequent use, they are somewhat subjective, as they do not collect data straight from the animals but from their owners [10]. In addition, observational studies bring some challenges, including confounding and recall bias, which may influence the quality of the data. The significant female response bias (90.6%) is well-recognised in both veterinary and human surveys. This may be due to the increased willingness of women to express opinions via a survey, female caregiver status, greater engagement in social media, or wider exposure to survey recruitment strategies [11].

A limitation of the study design is the lack of a control group of non-air travelling dogs to distinguish whether findings are specific to air travel or can occur during other stressful events or, in general, daily canine life. A further limitation of the study design is that the questionnaire was only available in English and, if the study were to be repeated for either dogs or other species, translation of the questionnaire into other languages, such as Spanish, should be considered to reach a wider and possibly more representative audience.

### 4.2. Demographic Data—Dog

The dogs in this survey are mostly “owned” companions or “pets”, likely have one or two dedicated caretakers, and are considered part of the family. This is validated by the fact that most dogs in this survey were adults or seniors (69.8%) at the time of air travel, and the two primary purposes for air travel were relocation (43.5%) and taking the dog on vacation (24.3%). Moreover, 80% of participants stated that their dog was obtained for companionship and 23.5% for different types of dog sports, indicating a strong bond. A strong human–animal bond may benefit animal welfare [12], and animals that are part of a human-animal bond are more likely to be treated as individuals and with special concern for their welfare [13], and it is, therefore, possible that the dogs in this survey were prepared for air travel with a greater level of concern for their welfare. However, a strong human-animal bond may also be the source of compromised welfare [12], and it is possible that, in some cases, air travel was chosen for dogs that were perhaps not suitable, either from a physical or a mental/emotional health perspective to undergo air travel. There is a second air travelling dog group, namely puppies or pregnant bitches, which are shipped either legally or illegally for the purposes of “puppy-trafficking” or resale and are faced with different or additional challenges to their health and welfare [14].

A total of 64.4% of dogs in this survey were either medium or small-size dogs (between 5 kg and 25 kg). It is possible that there are more small- and medium-sized dogs in the general dog population or that these sizes are more suitable for air transport, both from a logistical and financial point of view, and that owners who saw themselves relocating or travelling by air in the future chose small- or medium-sized dogs as companions to facilitate this.

Only 8.3% of dogs in this survey were brachycephalic breeds, which, given current brachycephalic breed popularity [15], seems underrepresented and contrasts with the number of brachycephalic breed dogs shipped through London Heathrow Airport between 2012 and 2017, which was 26% of all dogs [16]. It is possible that more brachycephalic breeds were transported by air between 2012 and 2017 and that this has now reduced due to perceived higher risks of air travel as well as airline reluctance to transport these breeds. It is also possible that there are regional differences and that more brachycephalic breed dogs were transported into and out of the UK compared to globally. Finally, it is possible that the high number of brachycephalic breed dogs being shipped through the UK from 2012–2017 were indeed large numbers of brachycephalic puppies being shipped for resale due to the popularity of these breeds. One omission in the current survey questionnaire was a question regarding the Brachycephalic Obstructive Airway Syndrome (BOAS) assessment of brachycephalic breed dogs, whether this was performed, and whether this had any consequence on decision-making and their ability to travel by air.

The most commonly reported physical ill health presentations at the time of air travel were environmental and food allergies (13.2%); however, only 2.2% of dogs were taking anti-allergic medication at the time of air travel. A total of 7.6% of dogs were reported to have osteoarthritis or another musculoskeletal disease; however, only 1.9% were being treated with non-steroidal anti-inflammatory medications.

A total of 44.3% of dogs were reported to be suffering from one or more mental health diseases, but only 5.5% of dogs were being treated with psychotropic medications. We need to consider whether physical diseases were diagnosed and being managed by a veterinarian and whether mental/emotional ill health disorders were diagnosed and being managed by a veterinary behaviourist or veterinarian with a special interest in behavioural medicine, who would most likely have recommended medical treatment, or whether these conditions were self-diagnosed by the participant. In one study on canine food allergies [17], 60% of dog owners suspected that their dog had a food allergy before their veterinarian, and it may be possible that pet owners reported diseases in their dogs in this survey without veterinary diagnosis or treatment. The internet is the most widely used source for health information, and given the fact that most pet owners think of their pets as family members and are just as vested in their pets’ health and wellbeing as other family members or themselves, it is reasonable to assume that many pet owners are using the internet to help educate and guide them in making decisions about the health of their pet, which may lead to self-diagnosis [18]. Obtaining an accurate diagnosis and initiating veterinary-prescribed treatment of both physical and mental/emotional health disorders before air travel may be an important factor in managing a pet’s physical, mental, and emotional well-being and welfare during and after air travel.

Almost 60% of dogs in this survey had travelled by air once or twice. Canine air travel can be complex to organise, financially expensive, and, according to this survey, is either extremely or very stressful for over 40% of pet owners and these may be some of the reasons for the infrequency of air travel for most dogs in this study. Under 10% of dogs were frequent flyers with over 10 air travel events. Further exploration of the data are necessary to find correlations and will be part of a future paper.

Two further omissions in Section 1 (dog demographic factors) of the questionnaire were noted at the time of data analysis: (1) information regarding whether dogs had been groomed (shaved or clipped) before air travel to help with temperature regulation and (2) the question regarding the dog’s neuter status was addressed to the time of survey completion and not at the time of the air travel. If the survey were to be repeated, these questions may want to be changed or included.

### 4.3. Demographic Data—Owner

When looking at nationality and country of residence of the participants, around half were North American, around one quarter were European and the remaining one quarter of participants were divided between Middle East/Africa, Asia, Oceania (Australia, New Zealand, and Pacific countries), and Latin America. This correlates approximately with the regions of both flight departures and arrivals for the dogs travelling by air (45–52% North America, 22–31% Europe, 10% Middle East/Africa, and 5–8% Asia) in this survey. It is likely that canine, domestic air travel within North America is both easier and more affordable due to reduced requirements for documentation (import- and export certificates), quarantine, and complex veterinary preparation such as blood tests and parasite control. This may explain why 50% of participants in this survey were North American and around 50% of flights both originated and terminated in North America. It may also be that air travel is just a more commonly accepted way of domestic travel for people in North America and that dog owners find this a “normal”, easy, and acceptable way of taking their dogs on trips with them. The same may apply to a lesser degree within the European Union. Canine air travel to Australia, New Zealand, and some countries in Asia involves much more financial commitment and preparation, sometimes months in advance, in terms of blood testing and documentation, and requires a mandatory quarantine stay for all dogs arriving, and this may account for the reduced numbers of participants from these countries and travelling to these countries.

As mentioned, over 40% of participants were either very or extremely stressed, and a further 34% were somewhat stressed by the thought of their dog travelling by air. This is much higher than the participant-reported stress levels of the dogs themselves at various points of air travel and in the first month after air travel. There could be several reasons for this. Owners may experience a lack of control, which can be stressful for many people [19]. Another factor adding to this may be the lack of transparency when it comes to pet air travel, especially when dogs travel in the hold of the aircraft. Owners are forced to relinquish control once the dog is handed over to the pet shipper or airport staff, and they often do not know where their pets, which may be regarded as “family members”, will be kept before, during, and after the flight. When asked about the care of dogs travelling in the hold during air travel, 16.7% of owners stated that they did not know where their dogs were cared for during air travel, and most owners knew that this would not be in a dedicated airport animal lounge. Dedicated airport animal lounges are only available in a small number of international airports, including Frankfurt, Amsterdam, London Heathrow, and JFK, and this may be an area of consideration when thinking about welfare improvements in international pet air travel in the future.

Despite many owners reporting feeling stressed, the participants did not rate their dogs as showing levels of stress in the same numbers as themselves. A reason for the lower incidence of reported stress levels in dogs could be a lack of education when it comes to the recognition of canine fear, anxiety, and stress signs. For example, some dog owners may not recognise a dog “freezing” and being very still due to fear and may interpret this as the dog being calm. In contrast, they would recognise and be able to report more accurately on feelings of stress experienced by themselves. It is also possible that dog owners do not know where and how to access professional advice and support when it comes to preparing their dogs for air travel, which may contribute to feelings of stress and being overwhelmed. A total of 66.8% of participants planned and prepared their dog’s air travel themselves, without the support of a pet shipper, and over half of the participants did not seek any advice from professionals to prepare their dogs for air travel. Making professional support, advice, and education regarding canine air travel more widely and easily available may contribute to reducing dog owner stress.

Finally, there are a number of alarming reports in the media of dogs dying both in the cabin and the hold of aircraft during flights [20,21], which can understandably be seen as a large source of stress for dog owners whose dogs are about to travel by air. While these events are horrific, they are not the norm for pet air travel, and most pets arrive alive and well.

### 4.4. Data about the Dog’s Air Travel Process and Experience

#### 4.4.1. Logistics and Preparation for Air Travel

Over 60% of flights in this survey were direct flights. As mentioned in the introduction, studies looking at air travel in horses [5] noted sharp increases in heart rate and changes in behavioural activities, especially during the transitional stages, such as the aircraft ascending and descending. Although these data do not exist in dogs, it may still be reasonable to assume that multiple transit phases with multiple ascents and descents may be more stressful for dogs, and therefore, the most direct route of air travel should be chosen wherever possible.

Interestingly, there was an almost equal distribution of dogs travelling in the cabin and the hold of the aircraft in this study. In the authors’ experience, travel in a cabin is restricted to small dogs (under 8 kg weight) for most international flights, and therefore, it is possible that the large number of North American domestic flights, where medium- and large-sized dogs are also permitted to travel in the cabin may have influenced the more equal distribution of these data.

As discussed previously, two-thirds of participants planned their dog’s air travel journey by themselves, and over half did not seek any professional advice when preparing their dogs for air travel. Reasons for this may include cost, not knowing where or how to easily access advice and support, and the fact that the logistics of domestic flight planning within North America is straightforward and may not require support from professionals. Further data analysis is necessary to see correlations between route/region and preparation for air travel and will be included in a future paper.

Where advice was sought from professionals, most was sought from the airlines and IATA, with veterinarians only accounting for around 30% and pet shippers accounting for only around 16% of advice given. This means that there is a great opportunity for veterinarians and pet shippers to be more involved with further education, especially when it comes to safe stress management methods for pet air travel. A recent paper [22] describes a number of safe stress management options for air travel in cats, many of which can be transferred to dogs. Owners and pet care professionals, including pet shippers, may not be aware of these options, and further widespread education on this topic is necessary and should be encouraged and embraced by all professionals in the pet shipping industry.

The most widely used air travel preparation methods were crate familiarisation (67.4%), physical examination by a veterinarian (around 75%), and medical procedures such as vaccinations, blood tests, microchip placement, and parasite treatments (around 80%). One of the most reproducible methods for reducing a pet’s stress levels during air transport is for the pet to be acclimatised to the carrier and trained to use it at home [23]. In the home environment, the carrier can become a ‘safe’ zone and/or a place that the pet associates with pleasant experiences, such as feeding, so that on the day of travel, the pet is less likely to be concerned about being inside the carrier [22]. Enough time must, therefore, be allocated for familiarisation with the carrier. The veterinary examination and other veterinary procedures are usually legal requirements for air travel, which is likely why these numbers were high. In contrast to this, almost 80% of participants stated that they did not use any stress management products such as anxiolytics, supplements, or pheromones, and seven participants were actively advised not to use any stress management products.

There is a lot of controversy about medicating pets for air travel, and much of this stems from the traditional use of sedatives and tranquilisers such as acepromazine. Almost half of airline transport deaths in animals from 1990 to 1995 resulted from sedation [24]. According to the IATA (International Air Transport Association) Live Animal Regulations (LAR) [25], sedating animals either before or during a flight is deemed a considerable risk to the animal and is, therefore, not recommended.

Historically, acepromazine has been prescribed to alleviate stress during air transport. However, tranquilisation with acepromazine (0.5 mg/kg) does not affect the physiological or behavioural stress responses in dogs during air transport [3].

Acepromazine is a phenothiazine derivative that has minimal impact on the animal’s emotional state of fear and/or anxiety; in fact, the tranquillising effect is instead dependent on motor inhibition mechanisms [26]. Phenothiazines have poor anxiolytic properties, cause sedation, and may increase startle reactions to noise [27], making them a poor choice for the management of fear, anxiety, and stress during air travel. In the authors’ opinion, it is likely that the historical use of acepromazine and its unfavourable effects have influenced opinion and the willingness to use other kinds of medication to manage stress in pets during air travel. With the advent of modern anxiolytics and other modalities for anxiolysis and stress reduction in the last years, it is important to consider anxiolytic medications to mitigate stress and, therefore, positively impact welfare in dogs travelling by air. What we should be considering now are medications that can be classed as “True Anxiolytics”, and that can alleviate fear whilst preserving the animal’s ability to function relatively normally, both emotionally and physically [28].

Short-acting anxiolytic medications that are commonly used for pet-related anxiety include gabapentin and pregabalin, trazodone, benzodiazepines, and maropitant for motion sickness [22]. There are some potential side effects to using anxiolytic medications, including (but not limited to) ataxia, sedation, and paradoxical hyperactivity [22], all of which are not desired during air travel and, therefore, anxiolytic medications should always be trialled before air travel and should always be used under the guidance of a qualified veterinarian familiar with these medications [22]. Effective anxiolytic supplements include those with the active ingredients alpha-casozepine (Zylkene^®^, Vetoquinol—Lure, France) [29] and L-Theanine (Anxitane^®^, Virbac – Carros, France, Composure^®^, VetriScience—VT, USA, Solliquin^®^, Nutramax—SC, USA) [30]. Canine appeasing pheromone products such as Adaptil^®^ (Ceva – Marseille, France), sprayed in all eight corners of the travel crate 10–15 min before putting the dog into the crate prior to travel [22], can also be very useful for managing stress during air travel [30,31,32,33].

#### 4.4.2. The Dog’s Experience of Air Travel

When looking at owner-reported canine distress levels before, during, and after the flight, an average of 10% of owners thought their dogs were very or extremely distressed at these times. Owners reported that they saw the highest number of very or extremely distressed dogs (13.1%) after the flight and the lowest number of very or extremely distressed dogs (6.1%) during the flight. The lowest number of distressed dogs overall during all 3 phases of air travel were reported to be those that travelled in the cabin with their caretakers, suggesting that being able to travel with your caretaker in the cabin is less stressful than travelling alone in the hold of the aircraft.

Body language signs of stress before, during, and after the flight were reported consistently, with panting, trembling, and whining being the three most commonly seen stress signs. It is possible that “panting” was a true stress sign; however, panting could also be a temperature regulation behaviour by the dog. The questionnaire did not specify whether “pawing” related to pawing at the crate door or possibly pawing at people, so it is difficult to know what this behaviour looked like exactly and whether it was an attempt to escape or get out of the crate or perhaps appeasement or attention-seeking behaviour. A further observation was that only a very small number of dogs urinated either during or after the flight, which may be due to a reluctance of well-house-trained dogs to urinate in an inappropriate location such as their crate or could be a sign of dehydration due to reduced water intake and panting. From a welfare point of view and considering that most of the flights described in this survey were long- (10–24 h) or medium-length (6–10 h) trips, providing better elimination opportunities during the process of air travel is something that should be looked at. Anecdotally, and from the authors’ experience at airports, there are not many suitable elimination areas that are hygienic or attractive to dogs, and this could be a suggested area of improvement for airports.

Around 10% of dogs were much more or extremely stressed than normal 48 h after the flight; however, this reduced over time to 0.6% 30 days after air travel. The number of dogs that were moderately stressed after air travel also reduced from 19.2% 48 h after air travel to 2% 30 days after air travel. The most common stress behaviours seen after air travel were “being more clingy”, being anxious, sleeping more, and being unsettled, all of which were reduced by 30 days after air travel.

One omission on the questionnaire in this section was a question about the state of repair of the dog’s travel crate upon arrival. This may have given further information about distress behaviours during the flight, such as pawing at or pushing into the crate door, as well as any traumatic experiences the dog may have had during the flight due to the crate being damaged externally, for example, from it being dropped or crushed.

The stress and distress behaviours evaluated in this survey were all owner or caretaker-reported behaviours, which poses some problems. Dog owners are often not educated in canine body language and may have overlooked or misinterpreted some behaviours. Again, dogs may have been “frozen” or still as an expression of their fear or stress, which may have been interpreted by the owner as the dog being “calm”, and this group of stressed dogs may therefore have been overlooked.

On the other hand, most dog owners know their dogs well, especially if they have been bonded for many years, and being in a stressful situation may make them more vigilant and observant of their dog’s behaviour.

Most dogs (85–90%) did not develop or did not experience a worsening of owner–reported behaviour problems after air travel. Of the dogs that did, the largest number showed signs of separation anxiety or showed more than one behavioural presentation. As mentioned previously, the owner-reported separation anxiety might not be a diagnosis in the true sense of the classification used in veterinary behavioural medicine but may rather be an expression of hyper-attachment shown by the dog after a stressful event. Hyper-attachment is characterised by the dog constantly looking for contact with their owners and being more likely to follow their owners around [34], a behaviour that may be consistent with the description of “more clingy” in the questionnaire. Hyper-attachment can result from several mechanisms including neoteny, which is the retention of infantile characteristics into adulthood and is a consequence of the domestication process and may have increased the tendency of some dogs to develop a strong attachment to their owners [34]. The development or worsening of more than one behavioural sign may be explained by the fact that stress affects overall mental health and increases the risk of generalised anxiety, separation anxiety, phobias, compulsive behaviours, and posttraumatic stress disorders [35].

Over 90% of dogs did not experience a worsening of owner-reported physical health signs after air travel, and of those that did, allergies and osteoarthritis/musculoskeletal problems were the most common presentations.

In all instances, dog owners felt that the development or worsening of behavioural or physical conditions was as likely to be attributed to changes in the physical or social environment as to the flight itself.

Sadly, two dogs in this survey passed away: one dog during and one directly after air travel in the quarantine facility. Although this is too small a number of casualties to detect any statistical significance, there are some characteristics that are noteworthy when considering these dogs. One dog was a brachycephalic breed dog, and the other dog suffered from owner-reported pre-existing mental/emotional illnesses, namely separation anxiety and fears and phobias of specific triggers, which were not being treated. Neither dog received any stress management products for air travel, and one dog was not familiarised with the travel crate. Further research is essential to understand which dogs are at greatest risk of losing their lives during air travel, and stakeholders must work together to prevent this.

The results in this final section of the questionnaire fit with our hypothesis that most dogs cope well with and recover quickly after air travel but that there are a certain number that suffer physical, mental, and emotional ill health consequences during and after air travel.

Finally, it is important to consider the limitations of this study and that the findings are based on owner-reported data, which may be biased or inaccurate. In addition, the number of participants in the study was relatively small compared to the number of dogs that travel by air every year, and data may, therefore, not be representative of all dogs that travel by air.

## 5. Conclusions

The results of this survey suggest that most owned pet dogs that undergo air travel cope with and recover well from air travel, but that there is a group of individuals who suffer physical, mental, and emotional ill health consequences during or after air travel, including death. Travelling in the cabin with the owner or a caretaker seems to be less stressful than travelling in the hold of the aircraft, as rated by the owners.

Most participants in this survey planned their dog’s air travel themselves, and over half did not seek professional advice for their dog’s preparation. Most dogs did not receive stress management products such as anxiolytic medication, supplements, or pheromones, and some owners were actively discouraged from using these. Of the professionals that advice was sought from, only 30% were veterinarians, and 16% were pet shippers, highlighting opportunities for these two groups of professionals, in particular, to better support and advise pet owners before their dog’s air travel and improve welfare outcomes for canine travellers. This survey also highlights improvements that can be made at airports, such as better provision for toileting areas and dedicated airport lounges for pets.

## Figures and Tables

**Figure 1 animals-13-03093-f001:**
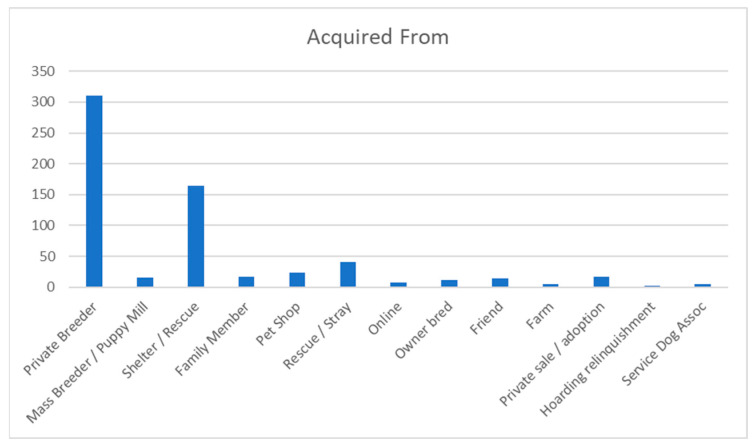
Where dogs were acquired.

**Figure 2 animals-13-03093-f002:**
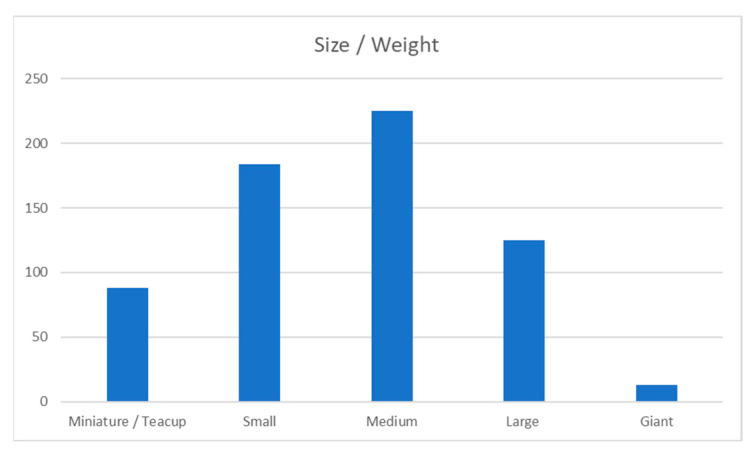
Size/Weight of dogs.

**Figure 3 animals-13-03093-f003:**
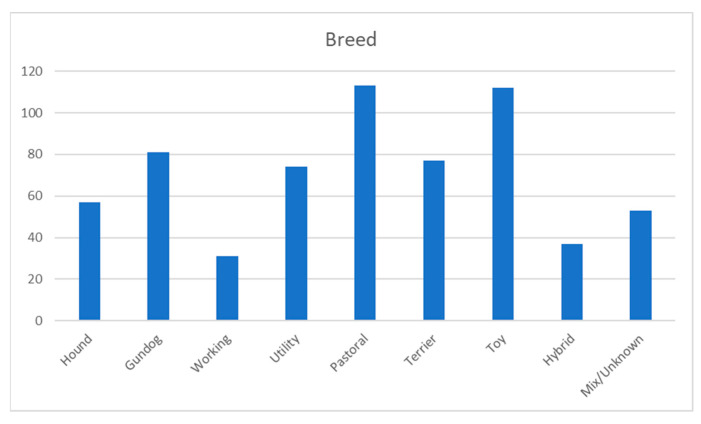
Distribution of breeds (according to UK Kennel Club breed groups).

**Figure 4 animals-13-03093-f004:**
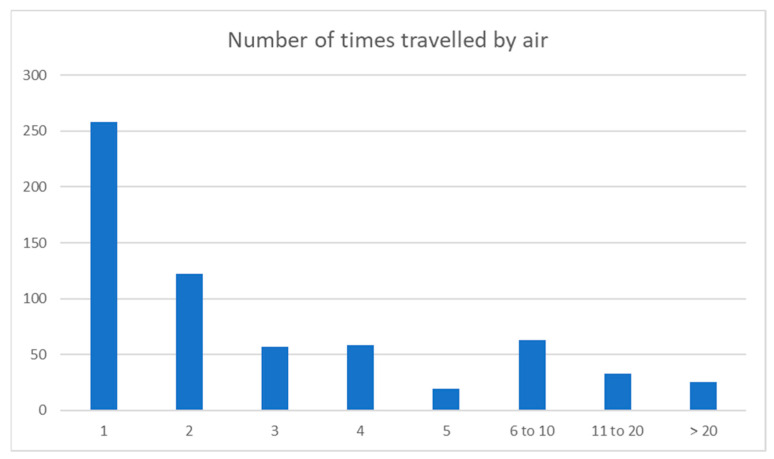
Number of times travelled by air.

**Figure 5 animals-13-03093-f005:**
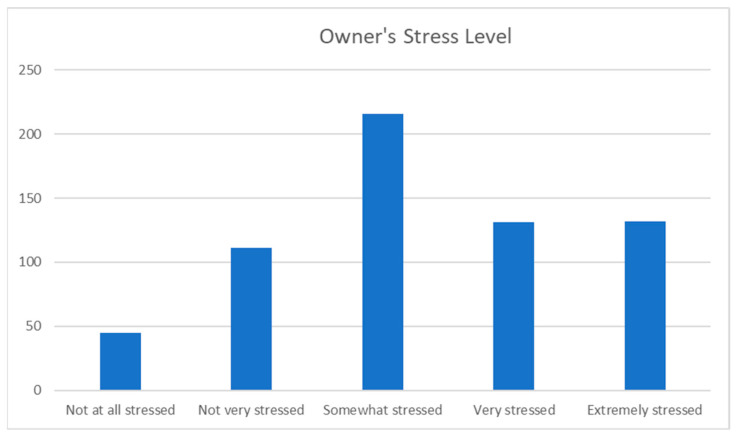
Owner’s stress level at the thought of their dog travelling by air.

**Figure 6 animals-13-03093-f006:**
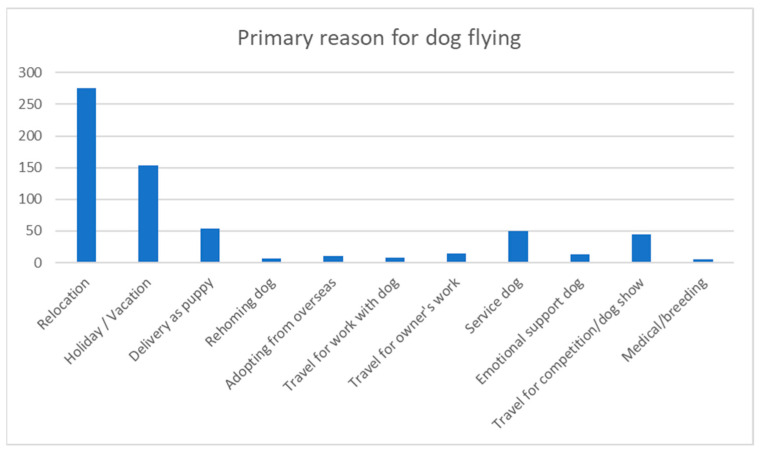
Primary reason for the dog flying.

**Figure 7 animals-13-03093-f007:**
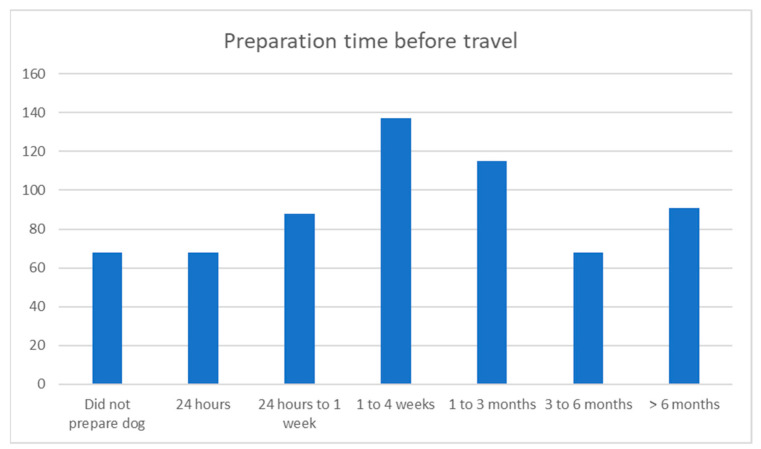
Preparation time before air travel.

**Table 1 animals-13-03093-t001:** Dog’s age at time of air travel.

Age	N	%
Under 6 months	64	10.1
6–24 months	128	20.2
2–4 years	146	23
4–6 years	98	15.4
6–8 years	95	15
8–12 years	63	9.9
Over 12 years	41	6.5

**Table 2 animals-13-03093-t002:** Owner-reported physical illness at time of air travel.

Illness	N	%
None	428	67.4
Allergies (environmental and food)	84	13.2
Osteoarthritis and musculoskeletal	48	7.6
Gastro-intestinal	17	2.7
Heart	13	2
Sensory impairment	10	1.6
Endocrine/liver/infectious	7	1.1
Respiratory tract	5	0.8
Neurological	4	0.6
Kidney	1	0.2
Neoplasia	1	0.2
3 or more concurrent conditions	17	2.7

**Table 3 animals-13-03093-t003:** Owner–reported mental/emotional illness at time of air travel.

Illness	N	%
None	354	55.7
Separation anxiety	93	14.6
Noise phobia	24	3.8
Fear/phobia other	37	5.8
Aggressive behaviours	19	3
Anxiety disorder	18	2.8
Compulsive disorder	4	0.6
More than one behaviour disorder	86	13.5

**Table 4 animals-13-03093-t004:** Advice sought from professionals (multiple answers possible).

Type of Professional	N	%
None	351	55.3
Airline	251	39.5
IATA	237	37.3
Books	202	31.8
Veterinarian	189	29.8
Internet/google	158	24.9
Pet Shipper	101	15.9
IPATA/ATA	61	9.6
Trainer	47	7.4
Family/Friends	41	6.5
Own experience	41	6.5
Social media	30	4.7
Breeder	13	2
Government agencies	4	0.6
Other pet professionals	116	18.3

**Table 5 animals-13-03093-t005:** Dog distress levels at various points of air travel.

Distress Level	Before Flight	At Airport or during Flight	Arrival/Collection/Delivery after Flight
Not at all distressed	26%	20.5%	30.4%
Not so distressed	22.7%	18.9%	23.5%
Moderately distressed	23.8%	10.7%	19.2%
Very or extremely distressed	10.2%	6.1%	13.1%
N/A	17.3%	43%	15.1%

**Table 6 animals-13-03093-t006:** Dog body language signs of stress at various points of air travel.

Body Language Signs of Stress	Before Flight	At Airport or during Flight	Arrival/Collection/Delivery after Flight
Panting	14.2%	13.1%	17.6%
Trembling	13.1%	9.4%	11.5%
Whining	11.5%	8.8%	14.5%
Tense	10.4%	7.2%	9%
Pawing	6%	7.7%	9.8%
Barking	6.9%	3.8%	8.8%
Cowering	4.6%	2.2%	5%
Hiding	5.8%	3.3%	5.5%
Pacing	3.6%	2.5%	6%
Urination	0.6%	1.7%	5%

**Table 7 animals-13-03093-t007:** Dog stress levels at various times after the flight.

Stress Level	48 h after Air Travel	3–7 Days after Air Travel	8–30 Days after Air Travel	Over 30 Days after Air Travel
Not at all or not much more stressed than usual	71.4%	84.4%	91.8%	96.4%
Moderately more stressed than usual	19.2%	10.4%	6%	3%
Much more or extremely more stressed than usual	9.4%	5.2%	2.2%	0.6%

**Table 8 animals-13-03093-t008:** Behaviours and stress signs seen at various times after the flight.

Stress Behaviours	48 h after Air Travel	3–7 Days after Air Travel	8–30 Days after Air Travel	Over 30 Days after Air Travel
More clingy	31.8%	15.4%	6.1%	3.5%
Anxious	18.1%	9.1%	4.4%	2.7%
Sleeping more	17.3%	-	-	-
Unsettled	14%	5.8%	-	-
Low appetite	13.9%	-	-	-
Less interactive	13.2%	-	-	-
Thirsty	11.7%	-	-	-
House soiling	7.4%	-	-	-
Sleeping less/poor quality sleep	6.8%	-	-	-
Fearful of triggers in the environment	6.6%	5.5%	3%	2%

**Table 9 animals-13-03093-t009:** Development of an owner–reported behaviour problem within 3 months after the flight.

Behaviour Problem	N	%
None	545	86.2
Separation anxiety	27	4.3
Noise phobia	5	0.8
Increased anxiety	19	3
Fear of specific triggers	6	0.9
Aggressive behaviours	6	0.9
Compulsive disorder	1	0.2
More than one behaviour problem	23	3.6

**Table 10 animals-13-03093-t010:** Worsening of an owner–reported behaviour problem within 3 months after the flight.

Behaviour Problem	N	%
None	568	89.6
Separation anxiety	26	4.1
Noise phobia	1	0.2
Increased anxiety	6	0.9
Fear of specific triggers	2	0.3
Aggressive behaviours	7	1.1
Compulsive disorder	1	0.2
More than one behaviour problem	23	3.6

**Table 11 animals-13-03093-t011:** Worsening of an already existing owner–reported physical illness within 3 months after the flight.

Illness	N	%
None	582	91.7
Allergies (environmental and food)	17	2.7
Osteoarthritis and musculoskeletal	12	1.9
Gastro-intestinal	8	1.3
Heart	1	0.2
Sensory impairment	1	0.2
Endocrine/liver/infectious	4	0.6
Respiratory tract	4	0.6
Neurological	1	0.2
Kidney	1	0.2
Neoplasia	2	0.3
3 or more concurrent conditions	1	0.2

## Data Availability

Not applicable.

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
