# Peer review of "How Well Do Dogs Cope with Air Travel? An Owner-Reported Survey Study"

_animals, 2023, doi:10.3390/ani13193093_

Round 1

Reviewer 1 Report

Excellent paper. I would like  to see a similar article on horses as they are also transported long distances. 

One thing that could be expanded was the  use of pheromones. How could thy be used ? Sprayed on dog or sprayed on kennel?  Also : not all air companies give people the "crates" used for the dogs.  Crate training is then difficult. These could be questions for  further research. How could airlines improve air travel for dogs,

Reviewer 2 Report

1. What is the main question addressed by the research?

 The main question is what are the effects of air travel on the behavior  and to a lesser extent, to the health of a dog.
2. Do you consider the topic original or relevant in the field, and if
so, why?It is relevant because many dog are subjected tot ransport by air.
3. What does it add to the subject area compared with other published
material? There have been few other studies
4. What specific improvements could the authors consider regarding the
methodology? The authors  point out that they have only the owner's impressions of the effcts of travel not those of a veterinarian or other animal professional.
5. Are the conclusions consistent with the evidence and arguments
presented, and do they address the main question posed? Yes 
6. Are the references appropriate? Yes 

Comment more on cabin vs hold results.

The labels on the figures are hard to read. Perhaps a different font or size should be used.

Reviewer 3 Report

  • The abstract could discuss the limitations of the study in more detail. For example, the study was based on self-reported data from pet owners, which can be subject to bias. Additionally, the study did not include a control group of dogs that did not travel by air, so it is difficult to determine whether the findings are specific to air travel or are more generalizable to dogs in general.
  • Methodology: The authors could mention how many countries and how many languages the survey was available in.

  • Results: Much of the information provided in the results is irrelevant. For example, the information about the dog's place of origin is not relevant to the study's purpose, which is to understand how well dogs cope with air travel. The demographic information, such as the age, gender, and nationality of the dog owners, is also not relevant to the study's purpose.

    The authors could provide more analysis of the information about where each dog was acquired and the breed of the dog. For example, the authors could compare how dogs from different sources or breeds cope with air travel.
  •  

    The figures are blurry, pixelated, and difficult to read. This is likely due to the fact that they are screenshots of the online survey questionnaire. 
  • Provide more descriptive statistics. For example, instead of simply reporting the percentage of dogs that showed signs of stress at different points during air travel, you could also provide the mean and standard deviation of the stress scores. This would give readers a better understanding of the distribution of the data.

  • Discussion. The authors could discuss the potential risks and benefits of using anxiolytics to manage stress in dogs during air travel. For example, they could discuss the potential side effects of anxiolytics and the importance of using them under the supervision of a qualified veterinarian. In addition, authors could discuss other ways to reduce stress in dogs during air travel, in addition to using anxiolytics. For example, they could discuss the importance of crate training and providing dogs with familiar toys and bedding during travel. They could also discuss the importance of avoiding feeding dogs too close to their flight time and providing them with plenty of opportunities to exercise and relieve themselves before and after their flight.

  • The authors could discuss the limitations of the study. They could mention that the study was based on self-reported data from dog owners, which may be biased or inaccurate. They could also mention that the study was relatively small and may not be representative of all dogs that travel by air. 

  •  
